# Empowering the Medicinal Applications of Bisphosphonates by Unveiling their Synthesis Details

**DOI:** 10.3390/molecules25122821

**Published:** 2020-06-18

**Authors:** Jéssica S. Barbosa, Susana Santos Braga, Filipe A. Almeida Paz

**Affiliations:** 1CICECO–Aveiro Institute of Materials, Department of Chemistry, University of Aveiro, 3810-193 Aveiro, Portugal; 2LAQV-REQUIMTE, Department of Chemistry, University of Aveiro, 3810-193 Aveiro, Portugal

**Keywords:** bisphosphonates, pharmaceutical applications, research-driven studies, synthesis improvement, reactional solvents, *P-*reactants

## Abstract

Bisphosphonates (BPs), well-known medicinal compounds used for osteoporosis management, are currently the target of intensive research, from basic pre-formulation studies to more advanced stages of clinical practice. The high demand by the pharmaceutical industry inherently requires an easy, efficient and quick preparation of BPs. Current synthetic procedures are, however, still far from ideal. This work presents a comprehensive compilation of reports on the synthesis of the commercially available bisphosphonates that are pharmaceutical active ingredients. Current limitations to the conventional synthesis are assessed, and paths towards their improvement are described, either through the use of alternative solvents and/or by selecting appropriate ratios of the reactants. Innovative processes, such as microwave-assisted synthesis, are presented as more environmental-friendly and effective methods. The main advantages and setbacks of all syntheses are provided as a way to clarify and promote the development of simpler and improved procedures. Only in this way one will be able to efficiently respond to the future high demand of BPs, mostly due to the increase in life span in occidental countries.

## 1. Introduction

Bisphosphonates (BPs) are a class of compounds with growing interest in the pharmaceutical industry. Their therapeutic properties in bone related disorders (e.g., osteoporosis, Paget′s disease or bone metastases) makes them the focus of several research studies as well as the main component of numerous pharmaceutical formulations [1,2]. There is, therefore, a high demand for synthetic processes that allow for their easy and rapid preparation.

The synthesis of BPs is widely described in the literature, with numerous reports presenting procedures with completely different reaction conditions [3,4,5,6]. Nonetheless, as it usually happens during the organic synthesis to prepare active pharmaceutical ingredients (APIs), in the pharmaceutical manufacturing the procedures often involve harsh synthetic conditions, characterized by the use of: (i) large amounts of solvents; (ii) long reaction times, with frequent multi-step reactions and involving extreme temperature and pressure conditions; (iii) the formation and subsequent disposal of waste and, sometimes, toxic residues [7,8,9]. In an era where the concepts of sustainability and sustainable development are in everyone’s mind, there has been an effort to develop simple, easy and safe processes to prepare BPs [3,5,10,11]. The current challenge is, thus, to adapt the synthetic procedures by making them safer and with lower impact in the environment, while keeping the affordability and efficiency demonstrated in the laboratories [8,9,12,13]. To better understand how this can be achieved, this perspective/review systematizes the current methods to prepare BPs and presents a critical discussion of the reaction parameters and conditions. The last segment of the paper provides an overview of the synthetic methods available within the scope of sustainable chemistry. Only in this way the industry and basic research at a laboratory scale will be able to efficiently respond to the high demand of BPs.

## 2. Bisphosphonate Applications

Bisphosphonates (BPs) were first discovered in 1894 by the pharmacist Theodor Salzer. At that time, they were mainly used in industrial processes (in the textile, fertilizer and oil industries) as corrosion inhibitors or as complexing agents [14,15]. The biological interest of BPs was only discovered after some studies of calcification mechanisms with inorganic pyrophosphate (PPi). It was shown that PPi, present in body fluids, could prevent the calcification of soft tissues by binding to hydroxyapatite (HAP) and avoiding its dissolution. Based on these interesting results, and because BPs are analogues of PPi (Figure 1), further studies were made to assess possible medical applications of BPs. In fact, it was shown that these compounds were able to inhibit soft tissue calcification and strongly inhibit bone resorption [16,17,18]. In their molecular structure, the presence of a P-C-P bond in the core embodies them with the ability to strong and selectively bind to HAP that composes bone tissue [2,15,19]. Upon adsorption, BPs have a catalytic effect over osteoclasts following different mechanisms of action for the non-nitrogen-containing BPs (non-*N*-BPs, etidronic and clodronic acids–the first-generation BPs) and the nitrogen-containing BPs (*N*-BPs, pamidronic, alendronic and ibandronic acids—the second-generation BPs and risedronic and zoledronic acids—the third-generation BPs) (Figure 1). In the former case, non-*N*-BPs are incorporated into adenosine monophosphate (AMP) originating toxic, non-hydrolysable adenosine triphosphate (ATP) analogues, while in the latter, besides the formation of the ATP analogues, the *N*-BPs inhibit the enzymes that synthesize key metabolites for the bone-dissolving osteoclasts. Ultimately there is an inhibition of bone resorption [15,20,21,22]. More than a century later, these structural characteristics and therapeutic properties are still the main focus for the wide application of BPs.

### 2.1. Worldwide use in Osteoporosis Treatment

Despite their solid and robust appearance, bones are dynamic tissues that undergo a constant remodeling process, crucial for their repair and renewal. This process consists mainly in two parts: (1) bone formation (by osteoblasts) and (2) bone resorption (by osteoclasts) [23,24,25]. In a healthy bone, the remodeling process involves the coupled action of osteoblasts and osteoclasts. In some cases, however, there is an imbalance between these two actions resulting in bone disorders, such as osteoporosis [26,27]. This disease is characterized by an exacerbated activity of osteoclasts and, thus, a higher bone resorption, that is not balanced by bone formation. As a result, there is a progressive loss of bone density along with a disruption of bone architecture and the concomitant increased risk of bone fractures (Figure 2).

In the European Union, in 2010, about 22 million women and 5.6 million men were affected with osteoporosis. Because the prevalence of this disease increases with age, and considering the growing life expectancy of the population, most likely we shall see an increase in the numbers of diagnosed people [28,29]. Besides the burden to the patients, there will be an increased stress on the economy, with direct costs from healthcare treatments and indirect costs that arise from loss of productivity of both patients and caregivers [28,29,30].

Nowadays, within the available and approved treatments for osteoporosis, BPs are one of the most commonly used class of drugs [28]. This may be, in part, because of their remarkably high affinity to bones, which enables them to achieve high concentrations in these tissues and, thus, a higher therapeutic effect when compared to other drugs that are non bone-specific [28,32]. Moreover, as BPs formulation patents expired, generic formulations became widely available, with these drugs becoming much more affordable and of widespread use [28]. There is nowadays a wide variety of BP-based pharmaceutical formulations available in the global market for the treatment of osteoporosis (which includes generic formulations), examples being presented in Table 1.

### 2.2. Other Pharmaceutical Uses

The medicinal applications of BP go beyond the osteoporosis treatment. In Europe, BPs began to be used in the early 1990s to treat cases of hypercalcemia (excessively high levels of calcium in blood) and to prevent changes in bone tissue in cancer patients. Only later, in the mid-90s, BPs began to be prescribed to patients with osteoporosis or Paget′s disease (pathology characterized by an exacerbated activity of osteoblasts and osteoclasts in some areas of the bone tissue, that, as a result, increases significantly, but presents a fragile and abnormal structure) [28,41,42]. In the USA BPs are on the therapeutic category of *“agents for metabolic diseases of bone tissue”* [43]. As so, in addition to their use in the prevention and treatment of osteoporosis [44,45], BPs are used for the prevention and treatment of osteolytic lesions associated with myeloma [46], or even bone metastases associated with breast cancer [47].

### 2.3. Seeking Novel Uses for BPs

Several research groups have focused their research on the search for novel and innovative applications of BPs. As shown in the examples highlighted in this section, the physico-chemical, biological and medicinal properties of BPs are being explored to the fullest, either in niche applications such as periodontitis, or in innovative uses such as bone-targeting drug delivery agents. An illustrated summary of these applications is presented in the Figure 3.

Through the apoptosis of osteoclasts BPs can provide an efficient treatment for periodontitis. This disease, predominantly of bacterial origin, is characterized by a pathological inflammatory condition of the gum and bone support (periodontal tissues) surrounding the teeth. In patients with this pathology there is a progressive loss of bone around the teeth, which gradually becomes looser and may end up falling off. To help prevent such outcome a pilot study used gel-based alendronate to fill the cavities in the periodontal area of five patients. After six months, a significant inhibitory effect on bone resorption was observed, with a visible increase in bone formation [48].

The remarkable selectivity of BPs towards bone tissue gives them potential for novel applications. In some of these instances, BPs are no longer be used as simple drugs, but as bone-targeting agents for the delivery of other drugs to bone tissue. An example is the anti-inflammatory Diclofenac (DIC), commonly used in the treatment of chronic rheumatism of the joints. Repeated and frequent administration of DIC is required for a successful treatment, but it may cause gastrointestinal problems. To avoid such effects Hata et al. functionalized DIC with a generic BP (tetraethyl aminomethylene BP, *R*_1_ = H and *R*_2_ = NH_2_). In the DIC-BP compound, the P–C–P core of the BP acts as a probe to induce transportation of a greater quantity to the bone, allowing a reduced number of administrations. In parallel, conjugation to the BP prevents the distribution of DIC in other tissues where it would have undesirable side effects [49].

The high affinity of BPs for bone tissue was proven useful in the field of imaging. In 2008, Bhushan et al. prepared what was nominated as a HAP-binding molecule, designed to contain a BP moiety as well as a functional part that incorporates isotopes such as ^99m^Tc [50]. Based on the same principle, another group of authors prepared a new DOTA monoamide analogue, able to establish a complex with the isotope ^68^Ga, and containing a BP moiety in its structure [51,52,53]. This way, with the BP moiety directing the compounds to the bone, both molecules can be effectively used as bone contrasting agents. Moreover, besides marking the entire skeleton, BPs tend to accumulate in higher amounts in areas with a high rate of bone remodeling. Therefore, they can be used to distinguish bone metastases or microcalcifications resulting from cancer (Figure 3A).

The phosphonic acid groups that are present in the structure of every BP are an interesting structural feature that can be explored even further. Novel applications can be envisioned such those reported by Wang et al. in 2006, in which magnetic and biocompatible nanoparticles coated with BPs were prepared. The phosphonic groups of the BPs coating these nanoparticles are strong chelating agents, able to efficiently and selectively remove uranyl ions from water and blood (Figure 3B) [54]. This chelating approach opens way for a very innovative method of removal of metal ions from body fluids that, at certain dosages, could present a toxicological risk for humans.

## 3. Synthesis of Bisphosphonates

Over the past years, the escalating demand for BP-based drugs and interest in BP-related research, required that these compounds could be rapidly and easily produced. We can find nowadays numerous reports and patents that describe different synthetic approaches for the synthesis of the BPs, in a process that is generally depicted in Scheme 1. Most of them focus on the improvement of the conventional synthetic procedure by modifying some of its reactional parameters. BPs can be prepared by several methods: (1) alkylation of tetraethyl methylenebisphosphonate, followed by acidic hydrolysis; (2) Michaelis-Arbuzov reaction between an acyl halide and trialkyl phosphite, forming a dialkyl acylphosphonate, that is then added to a dialkyl phosphite (in neutral medium); (3) by adding an acyl halide to a mixture of trialkyl phosphite and dialkyl phosphite; or others [55,56,57,58,59].

In the pharmaceutical industry, however, the aforementioned procedures are not commonly used. Instead, several studies and pharmaceutical patents described the synthesis of BPs as the reaction between an appropriate carboxylic acid (RCOOH), phosphorous acid (H_3_PO_3_) and phosphorous trichloride (PCl_3_), followed by a step of hydrolysis (Scheme 1, reaction 4) [55,56,60]. In some procedures, in alternative to phosphorous trichloride, phosphorous oxychloride or phosphorous pentachloride are used [61,62,63]. The BP obtained in the end of reaction 4 can be in its acidic form or the corresponding salt, depending on pH adjustments of the reaction mixture, in the final steps.

Reaction 4 in Scheme 1 is the focus of the current review because it is the foundation of the current synthetic procedures to prepare BPs in a pharmaceutical context, using as starting chemical common carboxylic acid molecules. Other previously mentioned procedures, though relevant from a synthetic chemistry perspective, fail to find a direct transposition into industry. Through the years a number of efforts have been made to simplify and improve the efficiency of reaction 4 in Scheme 1, either through the use of different solvents or by adjusting the quantities of *P-*reactants (phosphorous acid and phosphorous trichloride).

### 3.1. Solvent Selection

Reaction 4 (Scheme 1) is synthetically attractive: it tends to provide moderate-to-good yields, requires inexpensive and readily available reactants and involves a simple procedure, with a few simple steps. Depending on the solvent, however, the reaction mixture tends to be heterogeneous. A partial thickening of the reaction medium is sometimes observed, with the concomitant formation of a semisolid that inherently limits the ability to homogenize the reaction medium. As a result, there may be an incomplete reaction of the carboxylic acid and the reaction efficiency may be compromised [55,64,65,66,67,68,69]. There is currently a wide variety of solvents reported for reaction 4 as summarized in Table 2. The main aim is to use solvents that afford a homogeneous and fluid reaction mixture, thus improving the synthetic procedure.

#### 3.1.1. Chlorobenzene

The use of chlorobenzene as a solvent is patented for the preparation of risedronic acid [69], pamidronic acid [4,70] and alendronic acid [67] in moderate yields. An advantage of this solvent is the easy removal from the reaction mixture by distillation, with further possibility of reuse [6]. This is particularly relevant because these processes use up to 1 L of solvent. Nonetheless, it still remains a serious risk considering its potential hazardousness for the environment and humans (e.g., irritation in the eyes and skin) [68,69,70,71]. It is important to mention that chlorobenzene does not tend to solubilize the reaction mixture and thus, when used, a thickening of the medium with formation of a semisolid is typically observed in the reaction vessel [55,67].

#### 3.1.2. Methanesulfonic Acid (MSA)

Kieczykowski et al. studied the effect of various solvents. Using ether, THF, DMF, dioxane and hexane a thickening of the reaction mixture was observed, as in initial studies while using chlorobenzene. Only when using methanesulfonic acid as a solvent it was possible to keep the reaction mixture fluid, inherently allowing a higher conversion of the carboxylic acid into the desired BP [55,87].

Follow-up studies expanded the usage of MSA to the preparation of the BPs etidronic, pamidronic, alendronic, ibandronic, risedronic and zoledronic acids, or salts thereof [3,4,10,11,55,72,73]. Despite the overall good results, it is important to mention that MSA may not be adequate for industrial-scale production: it is expensive and, being an inorganic acid, it is toxic with corrosive and irritant effects [61,68,69]. When the desired product is the BP in the salt form, an additional step involving pH adjustment is required, which in this case may involve the formation of sodium methanesulfonate as an undesirable, difficult to remove bi-product [6]. It should be noted that the reaction between MSA and phosphorous trichloride is exothermic, resulting in significant heating of the reaction mixture [55,62,64,68]. To avoid this, lower reaction temperatures can be used, inherently resulting in longer reaction times that can reach days [55]. An alternative is to lower the reaction temperature before the addition of phosphorous trichloride, then blend it in very slowly and finally raise the temperature to values ranging between 60 and 80 °C [65,73,74,88].

#### 3.1.3. Sulfolane

The use of sulfolane in the reaction is reported to afford sodium pamidronate, sodium alendronate, disodium ibandronate, risedronic acid and zoledronic acid in moderate-to-high yields [10,68,72,74,84,89]. Even though this solvent has some inherent toxicity, it is a better alternative to MSA allowing safer procedures with non-reported uncontrollable exothermal reaction [6]. A few patents claim that this solvent allows a homogeneous and fluid reaction mixture [68,90].

#### 3.1.4. Polyalkylene Glycols

In recent studies, BPs were prepared in a mixture of solvents comprising a polyalkylene glycol (e.g., polyethylene glycol) and either a cyclic carbonate (e.g., propylene carbonate) [85] or toluene [77]. The mixture of toluene/polyglycol is reported to dissolve the reaction components, thus keeping the reaction mixture fluid and homogeneous. Note that this only happens when a co-solvent, such as toluene, is present [64,68,77,91]. Another important factor to consider before using polyglycols is their relatively high cost and their difficult removal [61,78,81,85]. This is even more important given the fact that they can be used in extremely high amounts, up to 800 g of polyglycols [77].

#### 3.1.5. Ionic Liquids (ILs)

Research groups worldwide have been focusing their work on the use of ionic liquids (ILs). Thus, it is not surprising that these acclaimed solvents have made their way into the synthesis of BPs. For instance, ILs consisting of salts of ammonium, sulphorium or phosphorium were reported as reaction media in the synthesis of pamidronic, alendronic, risedronic and zoledronic acids, or their respective sodium salt [78]. The reaction yields were low (ca. 31%) and little encouraging for industrial-scale use. Another study offered more promising results, with moderate yields of sodium alendronate (ca. 60%) using ILs such as [bmim][BF_4_], [bmim][Cl] or [bmim][PF_6_] (bmim stands for 1-butyl-3-methylimidazolium). This process was further improved when using sulfolane as a solvent and ILs as additives ([bmim][BF_4_] or [bmim][Cl]). It allowed to obtain moderate-to-high yields of sodium alendronate (from ca. 58% to 80%) [10]. In a series of similar studies, disodium ibandronate, risedronic acid and zoledronic acid were prepared in moderate to good yields using ILs such as [bmim][BF_4_], [bmim][PF_6_] or [bmim][Cl] [89]. During the synthesis of zoledronic acid, Nagy et al. noticed that the reaction yields seemed to depend on the amount of IL employed, in a curve peaking at 0.6 equivalents of IL (in respect to the carboxylic acid). At the curve maximum, zoledronic acid yield was 75%. When the IL was used as an additive and sulfolane as the solvent, the yield was increased to ca. 93% [84]. A similar outcome was observed during the synthesis of risedronic acid, with the highest yield (66%) being obtained using 0.6 equivalents of the IL [bmim][BF_4_]. When this quantity of IL was either decreased or increased (0.3 or 0.9 equivalents) the yield of the reaction lowered to ca. 51 and 58%, respectively [82]. In contrast, a study on the synthesis of disodium ibandronate with ILs demonstrated that the yield of the reaction appears optimal when using 0.1 equivalents of [bmim][BF_4_], affording 90% of the BP. When a 2:4 *P-*reactant ratio was used, the yield values were around 55%, regardless of the quantity of IL. Note also that, for disodium ibandronate, adding sulfolane to the IL brought no improvements to the reaction yields and rather lowers it to ca. 64–68% when using the same conditions that provided the above mentioned highest yield [79]. These curious results indicate that the synthesis of BPs in ILs warrants further optimization, tailoring the method to each desired BP. It must be highlighted, though, that ILs bring the advantage of being recyclable solvents [6,10]. This way, following the initial investment (because ILs are expensive), successive reaction cycles will make the process more economically interesting.

#### 3.1.6. Other Solvents

A myriad of other solvents was tested in the synthesis of BPs. For the preparation of alendronate and risedronate, several patents claim the use of non-ionic emulgators, such as alkanes, aralkyl or alkyl ethoxylates, or even triglycerides from plant or animal oils [64,83]. Claims on the synthesis of BPs include solvents as varied as phenol, *p*-cresol or *p*-nitrophenol [75,76], diphenyl ether [61], anisole [62], *N*,*N*′-dimethylethyleneurea (DMEU) [90], various types of hydrocarbons (halogenated or not), from linear to cyclic and aromatic, heptane, octane, cyclohexane, ethylene dichloride [86,88,92], toluene [63], decalin/methyl cyclohexane (alone or in a mixture, combined with orthophosphoric acid) [91], a silicone fluid (e.g., poly(dimethylsiloxane)) [63], or polar organic solvents (e.g., acetonitrile, benzonitrile, propionitrile, acetone, tetrahydrofuran or N-methyl-pyrrolidinone) [80,81].

A few of these solvents are claimed to avoid the thickening of the reaction mixture, keeping it fluid and, thus, allowing a higher efficiency [64,75,83,86]. Many of them are easily accessible and non-expensive. More importantly, they tend to be safer than those previously described (e.g., MSA), with no exothermal reaction occurring during the procedure [64,80,86]. Depending on the prepared BP, moderate-to-high yields can be obtained, from ca. 56% to 80% [61,62,63,64,75,80,86,91]. It should, however, be highlight that a few of these patents describe the use of the respective solvents in incredibly large amounts [63,64,76,83,86,91].

#### 3.1.7. General Remarks

BPs can be prepared using quite a few distinct solvents. Envisioning a future application in the pharmaceutical industry, one can strongly suggest the use of a solvent, or a mixture of solvents, with reported ability to ensure a homogeneous and fluid reaction mixture: MSA, sulfolane, cyclohexane, sunflower oil, phenols, triglycerides, a mixture of toluene/polyglycols or a mixture of sulfolane as a solvent with ILs as additives [3,4,10,11,55,64,68,72,73,75,77,83,86,87,90,91].

Results, systematized in Table 2, show that the yield tends to change depending on the employed solvent. Solvents such as sunflower oil and phenols, which afforded lower or non-reported yields, should not be considered [75,83]. It should be avoided the use of (i) cyclohexane and triglycerides, because incredibly large amounts of these solvents are needed to solubilize the reaction mixture [64,86], (ii) a mixture of toluene/polyglycols because of the possible contamination of the final product with polyglycols [61,77,78,81,85], and (iii) a mixture of sulfolane and ionic liquids as additives, at least not without conducting a preliminary studies to analyse the feasibility of scaling up the process [10].

From our literature survey we believe that sulfolane is the best choice as solvent. It is interesting to note that this solvent allowed the synthesis of several BPs in moderate-to-good yields, being safer than others employed in the literature [10,68,72,74]. We stress that the usage of MSA should apparently be solely considered for laboratorial scale processes. Though this solvent promotes moderate-to-good yields of BPs (while using the correct quantities of the reactants) [3,4,10,11,55,72,73], serious risks related to its use, as well as its hazardous properties, strongly discourage its selection for a production at an industrial scale [6,55,61,62,64,68,69].

### 3.2. Selection of the Molar Ratios of the P-Reactants

One of the main issues regarding the synthesis of BPs is choosing the correct amounts of *P-*reactants. From the first studies on this matter one could conclude that changing their quantities could result in different reaction yields, although the results were not properly systematized [55]. Recent studies demonstrated the effect of the ratio of the *P-*reactants on the overall yields. Relevant data is compiled in the following subsections.

#### 3.2.1. Synthesis in MSA

The influence of the ratios of the *P-*reactants on the synthesis of etidronate disodium, ibandronate sodium, risedronic acid and zoledronic acid was investigated using MSA as the reaction solvent (Table 3). Best yields (from ca. 38% to 83%) were obtained when only phosphorous trichloride was used [3,11,93,94], while large amounts of phosphorous acid typically led to lower yields. Using only phosphorous acid leads to no reaction [3,11,55].

#### 3.2.2. Synthesis in MSA vs. Sulfolane

The synthesis of pamidronic acid, or its corresponding sodium salt, was tested in MSA and sulfolane, while varying the ratio of the phosphorous-containing starting chemicals, as systematized in Table 4 [4,55,68,70,72]. In MSA and having only phosphorous trichloride as the phosphorous-containing starting chemical, a high yield was achieved, whereas the use of phosphorous acid alone did not allow the formation of the desired product [4,72]. When using both reactants, the yields tend to be lower as the amount of phosphorous acid increases [55,72]. Noteworthy, for a ratio of 2:1 or 2.2:1.1 (PCl_3_:H_3_PO_3_), moderate yields (ca. 57%, as obtained with only phosphorous trichloride) and low yields (ca. 28%) are observed, respectively [55,72]. It is, however, imperative to refer that the reaction conditions are different in these two cases, the higher yields being obtained with longer reaction times.

In sulfolane it is not possible to prepare the desired BP while using only one of the phosphorous-containing starting chemicals [72]. From the available data, the optimal ratios for these reactants cannot be ascertained. Results are non-linear because higher yields are reported for equivalent amounts of the reactants (2:2) and for a mixture with high content in phosphorous trichloride (3.4:1.5) [68,72].

#### 3.2.3. Synthesis in MSA vs. Sulfolane vs. Ionic Liquids vs. Sulfolane & Ionic Liquids

The synthesis of sodium alendronate was investigated in several solvents for a myriad of molar ratios of the *P-*reactants. Data is systematized in Table 5. With MSA as the reaction solvent and having solely phosphorous trichloride as the *P-*reactant, moderate to high yields of alendronate were obtained, in line with previous reports (of ca. 50% to 70%) [4,10]. On the other hand, using both reactants can either result in low (20%), moderate (60%) or high yields (90%) [10,55,73]. The difference between lower and moderate yields may be due to the different molar ratios used, of 1:2 and 3:2 (for PCl_3_:H_3_PO_3_), respectively. 

As previously observed, the yield of the reaction tends to increase as the quantity of phosphorous trichloride increases. Regardless of the impressive high yields (ca. 90%) obtained in a few studies, it is important to notice that in these cases the reaction time was significantly longer [55,73]. The fact that the reaction conditions (time and temperature) are different on the various studies available in the literature does not allow for a proper systematization of results nor for the establishment of a ratio-to-yield direct correlation. This subjective matter is even mentioned by some authors: for example, according to Nagy et al. [10], the procedure reported by Kieczykowski et al. does not give reproducible yields, as they obtained much lower yields than those previously reported (ca. 35% to 43%). This inconsistency between reports is observed while using only phosphorous acid as the phosphorous containing reactant. Either no reaction takes place (as in previous reports with other BPs) [11,72] or surprisingly high yields are obtained (ca. 70%) [95]. There is no reasonable explanation for these completely different results because, in fact, the reaction conditions in both studies are practically the same. Further investigations, including details of the used experimental apparatus, should be performed. The preparation of alendronate sodium in sulfolane while using different ratios of the *P-*reactants was investigated. When only phosphorous trichloride or phosphorous acid were used, yields were very poor, or even no product was obtained in measurable amounts. Higher yields were obtained when combinations of these reactants, at different ratios, were employed; more preferably when these were used in large excess in regard to the carboxylic acid [10,68,74]. The addiction of ILs as additives to this solvent afforded high yields as well, with better results for combinations having more phosphorous trichloride than phosphorous acid [10].

The effect of the *P-*reactants on the synthesis of zoledronic acid in distinct solvents is summarized in Table 6. With MSA the highest yields were obtained when employing solely phosphorous trichloride; a surprisingly high yield of 83% was reported in comparison with other moderate yields of ca. 50%. The reaction conditions that led to the best yield are characterized by the use of high amounts of the *P-*reactants and a long reaction time. Adding phosphorous acid to the reaction tended to lower the yield and, when only phosphorous acid was used, no product was obtained. Conversely, the reaction outcome was different when sulfolane was used as solvent. 

Using only phosphorous trichloride afforded very low yields of ca. 10%, whereas yield values increased significantly when this *P-*reactant was used in tandem with phosphorous acid. The highest yields, around 74–75%, were obtained for PCl_3_:H_3_PO_3_ ratios of 2:2 or 3:3. The use of the IL [bmim][BF_4_] as the reactional solvent led the yield to be seemingly independent from the *P-*reactants ratio, with no significant differences being observed for ratios of 2:2 or 3:2. A maximum yield of 74–75% was obtained for these ratios, while using 0.6 equivalents of IL. Moreover, while using these conditions of *P-*reactants and equivalents of IL, along with sulfolane as solvent, the authors were able to increase the yield of the reaction to ca. 93% and 91%. Once again, there was no significant difference on the yield of the reaction for the different ratios [3,68,84].

Risedronic acid was prepared with the highest yield (66%) when the IL [bmim][BF_4_] was used as solvent, in a quantity of 0.6 equivalents, along with a PCl_3_:H_3_PO_3_ ratio equal to 2:2. Changing the *P-*reactant ratio to 3:2 afforded a similar yield (65%). In sulfolane, the yields tended to be lower, with values of 58 and 49% for *P-*reactant ratios of 2:2 or 3:2, respectively. In this solvent, the highest and lowest yield values (60% and 2%) were obtained with a PCl_3_:H_3_PO_3_ ratio equal to 2:3 and 2:0. When sulfolane was combined with the IL [bmim][BF_4_], risedronic acid was obtained at similar yields as when using the IL alone, with no significant difference between the reactions carried with a *P-*reactant equal to 2:2 or 3:2 (65% or 67%) [82].

The aforementioned results of the syntheses of zoledronic and risedronic acids in ILs and mixtures of IL&Sulfolane showed no significant impact of the *P-*reactant ratio on the reaction yields. It should be noted, however, that only two ratios were investigated in those studies. A broader range of conditions was tested for the synthesis of disodium ibandronate (see Table 7). The authors tested different quantities of the IL [bmim][BF_4_]—0.6, 0.3 or 0.1 equivalents—, along with three ratios of PCl_3_:H_3_PO_3_, namely 3:2, 2:2 and 2:4. Regardless of the amount of IL used, yields were higher for the first *P-*reactant ratio and then gradually decreased following the order 3:2 > 2:2 > 2:4. Considering all conditions, the highest yield (90%) was obtained while using 0.1 equivalents of [bmim][BF_4_] and a PCl_3_:H_3_PO_3_ ratio equal to 3:2. This yield was even higher than that obtained in sulfolane with a PCl_3_:H_3_PO_3_ ratio of 3:4 (83%). In fact, with this solvent, the yields tended to be good or even high for equivalent amounts of *P-*reactants or when phosphorous acid was used in some excess. Reaction yield values lower with increasing amounts of phosphorous trichloride, reaching very poor values when it was used alone [79].

#### 3.2.4. General Remarks

The ratios of the *P-*reactants seem to be of great importance when the synthesis is carried in MSA. Yields in BPs increase with the increasing ratio of phosphorous trichloride to phosphorous acid. In fact, the use of this reactant alone usually did not afford a product, allowing one to postulate that it may be a spectator in the reaction. Nonetheless, when using different solvents this concept may not apply. As seen previously, when the synthesis takes place in sulfolane, using only one of the reactants results in poor yields. These results may be due to the particular ability of MSA to react with phosphorous trichloride and trigger the reaction to start [6]. As so, it seems that solvents unable to react with this compound, such as sulfolane, require the presence of both *P-*reactants in the reaction medium. On this note, the role of the *P-*reactants during the synthesis of BPs in ILs should be better elucidated. A broader range of ratios, including the use of only one of them at a time, should be employed to help understand their impact in the reaction yields.

### 3.3. Innovative Processes

In plain 21st century, worldwide research groups try to embrace the concept of sustainability in their studies, as a way to “reduce the environmental impact of processes and products, optimize the use of finite resources and minimize waste” [96]. It is important to develop synthetic procedures with less harsh experimental conditions (lower pressure and temperature conditions; less quantity and less hazardous solvents, to avoid unnecessary costs and risks for both human health and the environment) but allowing, at the same time, the synthesis of the desired compounds in high yields and with enhanced purity. One of the synthetic methods that have gained considerable interest over the past years is the Microwave-Assisted Synthesis (MWAS), already in use by many research groups in their organic synthesis processes [97,98,99,100].

MWAS is a simple and highly efficient process that, due to a uniform and faster heating of the reactants inside a microwave-transparent vessel, permits a considerable reduction of reaction times and increased reaction yields. Moreover, within the pharmaceutical industry, this synthetic approach is particularly appealing due to its selectivity, since it typically generates phase-pure products [97,101,102].

The use of MWAS was reported for the preparation of pamidronate sodium, alendronate sodium, risedronic acid and zoledronic acid by means of a two-step reaction synthesis (Scheme 2, reaction 5) [103]. Several BPs were obtained upon a total reaction time of ca. 17 min, which is significantly less when compared with the conventional heating procedures with reaction times taking several hours. The fact that the reaction yields are equivalent to those of conventional procedures (Table 8) makes these results all the more remarkable and relevant. Sulfolane could even be used as the solvent in the reaction mixtures.

Solvent-free reactions have been attracting the attention of several research groups because this procedure avoids the use of harmful organic solvents. In a recent study a library of BPs was prepared through MWAS, but without using any type of solvent. It was instead used silica gel as a solid support [104]. BPs were obtained by a two-step reaction method: (1) a mixture of RCOOH:PCl_3_:H_3_PO_3_ (in a ratio of 1:3:5) and silica gel was subjected to microwave irradiation for 3 min at 80 °C; then (2) the hydrolysis of the resulting reaction mixture was carried out in water, under microwave irradiation, for 3 min at 100 °C. In the end, monosodium salts of etidronate, pamidronate, alendronate, ibandronate, risedronate and zoledronate were obtained in high yields (78, 67, 78, 72, 86 and 80%, respectively) and in a pure form. The authors mentioned that when using conventional heating at 100 °C this type of synthesis in silica gel did not afford the desired BPs.

These two aforementioned studies illustrate how microwave irradiation allows for the rapid, simple and efficient synthesis of BPs. MWAS with silica gel allowed higher reactions yields for sodium salts of pamidronate, risedronate, zoledronate and especially alendronate. Because both studies have several different reaction parameters (e.g., molar ratio of reactants, time of irradiation and reaction temperature) it is not possible to immediately correlate the high yields with the use of silica gel instead of other solvent, such as sulfolane. This clearly shows that there is a significant lack of information in the literature and that further studies focused on the MWAS of BPs are clearly needed in order to better understand and improve the reactional process. We further stress that these procedures refer to small-scale syntheses. It is, thus, crucial to access their overall feasibility for scale-up approaches.

### 3.4. A Mechanistic Overview of the Synthesis

The outcome of BPs synthesis from reaction (4) is without any doubt, strongly dependent on two main parameters: the solvent used in the reaction and, the ratio of the *P-*reactants. Nonetheless, upon analysis, it seems that there are no optimal conditions at which BPs can be universally prepared at higher yields. Meaning that, depending on the applied solvent and, sometimes, even its quantity (evident in the case of ILs), the ratio of the *P-*reactants should always be optimized to find the precie proportions that allows improved yields.

This is evident when using two distinct methodologies where phosphorous trichloride is used as sole *P-*reactant majorly dissolved in MSA vs. a combination of phosphorous trichloride and phosphorous acid as *P-*reactants in general dissolved in Sulfolane. As mentioned earlier, these curious results may be due to the particular ability of MSA to react with phosphorous trichloride and initiate the reaction, which does not happen for other non-polar solvents, such as sulfolane, that requires both *P-*reactants [6]. To better understand this and clarify the role of each *P-*reactant, as well as the role of the solvent in reaction (4), a few possible mechanisms are suggested in literature [3,5,6,72,105,106,107]. Additionally, a detailed and critical analysis over several proposed mechanisms was presented in 2017, by Keglevich and co-workers [107]. Considering the various reports published in recent years, we describe below a refined mechanism for the synthesis of BPs in MSA or sulfolane.

A possible mechanism for the synthesis of BPs in MSA, using only phosphorous trichloride as *P-*reactant, is presented in Scheme 3A [6,72,105,107]. The first step, is the reaction of the carboxylic acid (RCOOH) with both: (i) phosphorous trichloride, forming the acid chloride (Path A), or with (ii) methanesulfonyl chloride (a product from the reaction of MSA and phosphorous trichloride), leading to the formation of a mixed anhydride intermediate (RCOO-MSA, Path B). This anhydride may also be formed upon the reaction of the acid chloride with MSA. The reaction continues with a nucleophilic attack to the carbonyl group of RCOO-MSA and/or acid chloride, by an anhydride intermediate Cl_2_P-O-SO_2_Me (formed by the reaction between MSA and phosphorous trichloride Scheme 3B). As a result, two adducts are formed and, upon the loss of MeSO^3−^ and Cl^−^, they are transformed into the corresponding keto intermediates. These stable forms can then react with another molecule of Cl_2_P-O-SO_2_Me, leading to the formation of the last intermediate. In the final steps, its hydrolysis allows to obtain the targeted BP.

On the other hand, when sulfolane is used as solvent, the presence of phosphorous trichloride and phosphorous acid is required for the synthesis of BPs. Here, the mechanistic aspects of reaction 4 are slightly different (Scheme 4A) with the initial formation of the acid chloride, as the sole nucleophile species [6,72,107]. Hereon the intermediates (HO)_2_P-O-PCl_2_ (Path A) and/or (HO)_2_P-O-PCl-O-P(OH)_2_ (Path B) attack the carbonyl group, originating the corresponding keto phosphonic acids. The intermediates are formed by the reaction of one (Scheme 4B) or two (Scheme 4C) molecules of phosphorous acid with one molecule of phosphorous trichloride. afterwards, another nucleophilic attack to the carbonyl group by (HO)_2_P-O-PCl_2_ intermediate allows the formation of a compound, that upon hydrolysis leads to the targeted BP.

As a final note, it seems that polar solvents, such as MSA, capaable of forming nucleophile species with phosphorous trichloride and of attacking the intermediate´s carbonyl group, do not require the presence of phosphorous acid in the reaction medium. In turn, other solvents, such as Sulfolane, imperatively require the presence of both *P-*reactants, so that they can react within them and form nucleophile species, able to react with the intermediates carbonyl group and proceed with the reaction. Regarding the mechanistic aspects of the reactions in ILs solvents, Nagy et al. suggested that they may increase the electrophilic character of the carbonyl group of the starting carboxylic acid and its chloride derivative [10,107]. Further studies are needed for a full elucidation of the role of ILs.

### 3.5. The Challenge of Reaction Workup to Achieve Pure Bisphosphonates

Reaction (4) in Scheme 1 represents only part of the whole process required for the preparation of pure BPs. The second step of reaction (4), the hydrolysis with water, does not immediately afford the BP in the pure state but rather a crude product. As so, additional steps are required to purify the product, in a time consuming and sometimes yield hampering process.

From our perspective, the processes of purification of the crude products are lacking uniformity. Indeed, literature reports present a vast collection of approaches for this process. In the simplest case, upon hydrolysis in water, the temperature of the reaction is lowered to precipitate the pure BP [72]. Nonetheless, in most of the reports, upon hydrolysis, there are additional steps of washing (with water) and/or digestion of the crude product (with ethanol, methanol, or a combination of these and water), involving one or more cycles. Additionally, pH adjustments of the reaction medium, may be used, particularly when the BP is to be isolated as a salt. Nonetheless, in a few instances, this is a one step process, with the pH being adjusted to values between ca. 4–6, but often it involves an initial pH adjustment to ca. 1.8, and a second pH adjustment to ca. 4–6 [10,55,61,63,72,73,75,76,79,80,84,85,95,105]. From the brief experience of our group in the laboratory, pH adjustments are, without any doubt, a crucial step to prepare BPs in the salt form, with the two-step process being the best approach. The washing and digesting steps are fundamental for the purity of the BPs. Surprisingly, a few reports in the literature present no purity values [55,61,85,95], or there is no mention of which technique was employed to determine the purity [75,76] (with the most common being nuclear magnetic resonance, but high performance liquid chromatography is also employed).

The variety of approaches and combinations thereof that can be used to treat the crude product mean that ultimately, the quantity and purity of BPs will also have a large variability. In fact, this may help explain some inconsistencies found in literature reports. Further efforts should be made to standardize these final steps, which should always be followed by information regarding the purity of the prepared BPs. Only in this way will the reports convey clear and coherent data that contributes to the evolution of synthesis processes of BPs.

## 4. Conclusions

BPs are compounds with a remarkable pharmaceutical interest, integrating a wide range of pharmaceutical formulations and commercial brands of medicines. These formulations can be found on the daily lives of many patients, and they are used for the treatment of pathologies in bone tissue and metabolism, with the most frequent being the treatment and prevention of osteoporosis, but including other pathologies such as Paget′s disease and some neoplasic bone lesions.

There has been an increasing interest in exploring novel applications for BPs. Based on their therapeutic properties and structure, there has been an attempt to efficiently broaden the use of these compounds. Within the new applications of BPs, their use to obtain bioconjugates, that can then be used as imaging agents for bone tissue stands out. The potential of this type of application is strongly driven by the high specificity of BPs towards bone tissue. Its eventual applicability is, however, conditioned to the toxicological safety of the radioisotopes that will be used in such conjugates.

To make all applications possible, both the ones that are already well-established and those that are still a vision, it is imperative to have efficient procedures for the synthesis of BPs that enable their continuous use. In this context, this work points out the main setbacks associated with the synthetic procedures of the BPs that are currently used as medicines. The difficulty in solubilizing the reaction mixture and in selecting the appropriate ratios of the *P-*reactants were identified as the main issues. Solutions to the first issue have been proposed by several research groups, with interesting results deriving from the use of several distinct solvents. In our perspective, sulfolane seems to be the solvent that gathers the best properties for further application in industrial-scale processes because it promotes good yield and purity, even when used in low amounts. Moreover, and despite some inherent toxicity, sulfolane can be tolerated in residual amounts (up to 160 ppm) in pharmaceutical products [108] and it is more environmentally and human-friendly than the second most well-known solvent, MSA. In fact, MSA is the subject of various patents on BP preparation, meaning that it is a functional solvent both for laboratory- and industrial-scale production. Providing that adequate caution is taken with the temperature of the reaction medium, MSA can also be used to afford BPs in good yields.

The *P-*reactants ratio in the synthesis of BPs seems to depend on the reaction solvent. While using sulfolane it is imperative to have two varieties of such reactants in the reaction mixture to ensure good yields of BPs. On the other hand, when the reaction takes place in MSA, the use of only phosphorous trichloride as reactant is enough to obtain BPs in good to moderate yields.

There is no doubt that both the solvents and the quantities of the *P-*reactants can affect the efficiency of the synthetic procedure to prepare BPs. In the end, one should select an appropriate and safe solvent (e.g., sulfolane), and the ratios of the reactants should be adjusted to the selected reaction conditions. Considering high efficiency and good results, more effort should be applied in the use innovative processes such as MWAS. Only in this way will the pharmaceutical industry be able to, little by little, find an effective and sustainable process for producing sufficient BPs to meet global needs.

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
