# Peer review of "Empowering the Medicinal Applications of Bisphosphonates by Unveiling their Synthesis Details"

_molecules, 2020, doi:10.3390/molecules25122821_

Round 1

Reviewer 1 Report

This is an excellent review article. Just a few suggestions:

1) it will be beneficial for the readers to include a short paragraph on aminophosphonates and aminophosphonic acids, the phosphorus-containing analogs of natural amino acids; some types are already shown in Figure 1, such as, would be number, 5, 6, 7, the omega-aminophosphonic acids; therefore the elaboration to more general field of aminophosphonic acids, including alpha-, would be quite natural and logical. 

2) the references to web-sites should be avoided as these web-sites can be taken down. In particular, the citation of FDA database seems the same for the number of references. I suggest to delete the reference number and include the web-address in the text. This should be the editorial policy of Molecules. The reference is used to support the statement, therefore it should be reliable and sustainable, not as fluid as a web-site. 

Author Response

We thank the reviewer for the positive comments on our manuscript and for the questions raised which are answered in the same order as appeared in the original review:

1) Aminophosphonates, bearing a single phosphonic group and being analogues to natural aminoacids, have a quite different range of pharmacological action, mainly involving antimicrobial activity and inhibition of the growth of some plants. Other aminophosphonates are toxic. For example, this is the case of the most controversial glyphosate.

To the best of our knowledge, no aminophophonate is approved as a medicine for humans. For this reason, we believe that this class of compounds is far outside the scope of the present manuscript.

2) Websites are mentioned in the MDPI reference style guide (page seven) and they should be quoted as references, rather than stay included in the text.

Reviewer 2 Report

The review paper is concerned with the synthesis (with the emphasis on the simple and optimized procedures) and medicinal applications of bisphosphonates. The paper will be of interest for synthetic chemists,  biochemists and especially for people dealing with biomedical investigations.

Although the paper is well written and organized, in the Section 3 Synthesis of Bisphosphonates” some recent relevant references concerning aminobisphosphonates, fluorinated aminobisphosphonates are not included, e.g.:

Molecules 2016, 21, 1474; doi:10.3390/molecules21111474; Tetrahedron 2014, 70, 2928-2937; Arkivoc 2012 (V), 127-166.

I suggest that the paper can be accepted for publication after consideration the above points.

Author Response

We thank the reviewer for the positive comments on our manuscript. Upon a careful analysis of the mentioned papers, we found that those pertaining to Molecules 2016, 21, 1474 and Arkivoc 2012 (V), pages 127-166 would be helpful to give the reader a broader view over the alternative synthetic strategies to prepare bisphosphonates, as well as to give them an understanding over the possibilities to functionalize these compounds for further use. We thank the reviewer for pointing them out and the new references were inserted in the end of the first paragraph in the chapter 3, Synthesis of Bisphosphonates (line 166).

Regarding the paper in Tetrahedron 2014, 70, pages 2928-2937, from our point of view, fluorinated aminobisphosphonates do not fit the purpose of the present review (very much in line with the comments for the first reviewer). We consider it as a class of distinct compounds from bisphosphonates, that are well-known APIs with biomedical applications.

Reviewer 3 Report

In their article, the Authors reviewed the syntheses of bisphosphonates focusing on the industrially relevant procedures, innovative novel approaches and applications.

This is a very well-written article which will generate great interest from chemist interested in this research topic. Besides the synthetic part, the paper also covers briefly the medical applications which is also nice overview.

There are several synthetic procedures for the preparation of BPOs, and the authors chose to focus on the procedure which is the most relevant for the pharmaceutical industry. Thus, they reviewed the synthesis of BPOs which involves the reaction of the corresponding carboxylic acid, phosphorous acid and phosphorous trichloride. The authors tried to systematize the results published in papers or patents. In this review, mainly the different solvents and molar ratios of the phosphorous-containing reactants are covered and systematized. In my opinion, tables containing the data extracted from the papers give a thorough overview of the literature data, and they are really informative, and they add great value to the paper.

One of my biggest concerns regarding this review is that the latest article or patent cited was published in 2017. Thus, this review does not cover the last three years of relevant bisphosphonate synthesis. The reviewer did a quick research on Web of Science and Scifinder using the following keywords in either the titles or the topics of the articles: ‘bisphosphonate synthesis’; ‘*dronic*’ and ‘synthesis’; ‘*dronate*’ and ‘synthesis’

This search revealed that there are several papers and reviews (!) which are in connections with this current review. I strongly recommend the authors to check the most current literature and incorporate those articles and patents which are relevant for this current review.

During the search the individual names of BPOs should also be used in the search engines instead of the general ones listed above. Moreover, I also recommend to do an author search which covers the experts and research groups who are active on this field (and they are cited multiple times in the current version) to look for their most recent papers which may be in connection with this current review.

In my opinion, a short section summarizing the mechanistic details of this reaction would add value to this review. The different mechanisms described by various authors / research groups should be summarized and compared in this new section. The Authors point out that the reaction parameters are inherently different in different article/patents, and it is a really demanding task to see through the data. This is the reason why the Reviewer thinks that a summary of proposed mechanisms would be of great interest, and this section may give new ideas to the Readers.

The Authors should devote at least a few paragraphs to summarize the literature from the following perspectives:

Workup of reaction BPO reaction mixtures

Purification of crude BPOs

Analytical techniques for purity check and quantification of API content

These topics are also quite relevant from industrial perspective. Moreover, the difference workup/purification/analysis also causes differences in the results, and in their interpretation. This is the reason why the Reviewer thinks that these key elements of BPO synthesis should also be addressed here.

Minor remarks:

Table 1: Does this table contains only the BPOs approved by FDA but not other agencies? If so, it should be stated in the text or in the title. The corresponding references indicate that FDA database was accessed in Jan 2018. The database should be checked again, and the table should be completed.

In Figure 1 and in the corresponding text, the different generations of BPOs should be marked.

The title of the tables needs improvement:

e.g. Table 3: Reaction parameters used to synthesize etidronate disodium, ibandronate sodium, risedronic acid and zoledronic acid in MSA through conventional heating (or using conventional heating) is a better word order.

The references should be checked for errors, as there are several issues. A few of them are listed below:

inconsistent capital letters: e.g. Tetrahedron letters or Tetrahedron Letters Ref 3 and 98

missing information: ref. 10

The use of DOI is inconsistent. In some references, it is shown, in other ones, it is not. Please check the guidelines of Molecules.

Non-English terms: e.g. acedido em 26-11-2019

Author Response

We thank the reviewer for the overall positive comments on our manuscript.

Upon revision, a handful of studies were added to the document, which inherently involved modifications in the text of the subsections 3.1.3. (line 228), 3.1.5. (line 249-269), 3.2.3. (line 357-404) and 3.2.4 (line 414-417), as well as changes in Table 2 (line 206) and the addition of Tables 6 (line 423) and 7 (line 425).

Following the suggestion of the reviewer, a short summary of the possible mechanisms behind the reactions to prepare bisphosphonates was added to the revised manuscript. Upon a revision of the literature, we found a review, from 2017, by Keglevich and co-workers, that meticulously summarizes and discusses several mechanisms that were proposed throughout the years. As so, we decided that the best option was to mention this review, for any reader to consult in case he/she wishes more scientific details on this matter. Nevertheless, in the current manuscript we describe solely the mechanism that is now the most acceptable (line 476-528).

In addition, the manuscript was also revised with topics included so that the reader would be aware of its implications in the synthesis of bisphosphonates (line 529-557). These were mentioned by the reviewer.

Answer to the minor remarks mentioned in the review:

  • Table 1 now contains formulations approved either by FDA or Infarmed, which is now explicit on the title. Both databases were checked and, as a result, some changes were made, either due to a discontinued product or due to a change in the manufacturer (line 102).
  • Figure 1 (line 70-71) and the corresponding text (line 60-63) were changed accordingly.

  • The titles of all tables were modified according to the recommendation (lines 206, 418, 419, 421, 423 and 425).

  •  All references were checked and corrected when needed.

Reviewer 4 Report

In their manuscript Barbosa, Braga and Paz  report about the innovative processes for the synthesis of bisphosphonates as important class of API to contrast osteoporosis and other bone related diseases. The approaches reviewed in the manuscript cover the use of alternative solvents and more efficient reaction conditions. Overall the manuscript presents an interesting point of view for the synthesis of an important class of API. The paper is well written, overall, I consider the manuscript suitable for publication after minor revision in accordance with the below reported comments:

-Form the current title and from the abstract and conclusions one would think that also several methods for BP synthesis in water should be reviewed which is not the case because those methods rely on more elaborated BP. I suggest to clearly state in the abstract and in the conclusions that the review is limited to commercially available BP API molecules.

 -The molecular structure of BPs in figure 1 should be better reported as PO(OH)2 rather than PO3H2

-Why BPO instead of simply BP for the acronym for bisphosphonates?

-There is a mistake in the structure of ibandronic acid (Figure 1), a missing H in the C atom in between the two P atoms

-Pay attention to uppercase and lowercase numbers in formula in all tables

Author Response

We thank the reviewer for the positive comments on our manuscript and for the questions raised which are answered in the same order as appeared in the original review:

  • Following the recommendations of the reviewer the following changes were made:
    Change in line 15: “Current synthetic procedures are, however, still far from ideal. This work presents a comprehensive compilation of reports on the synthesis of the commercial available bisphosphonates, that are pharmaceutical active ingredients.”
    Change in line 571: “In this context, this work points out the main setbacks to an improved and more efficient synthesis of these commercial available APIs”
  • Figure 1 was changed according to the comment (line 70). Other schemes were also changed to fit this recommendation (Scheme 1, Scheme 2) (line 183 and 449, respectively).

  • Upon considering the comment from the reviewer, the authors decided to change BPO to BP, probably a more well-known and used acronym for this class of compounds.

  • The structure of ibandronate was corrected (line 70). We thank the reviewer for pointing out this mistake.

  • Following the recommendations of the reviewer, changes were made in Table 4  (line 419).

Round 2

Reviewer 3 Report

The Authors changed many paragraphs of manuscript, and they answered all the questions. In my opinion the changes and the new text add great value to this nice paper. In my opinion, this paper will generate general interest from the readership.

I recommend the publication of this review.